# IMPROVING GREEDY CORE-SET CONFIGURATIONS FOR ACTIVE LEARNING WITH UNCERTAINTY-SCALED DISTANCES

## ABSTRACT

We scale perceived distances of the core-set algorithm by a factor of uncertainty and search for low-confidence configurations, finding significant improvements in sample efficiency across CIFAR10/100 and SVHN image classification, especially in larger acquisition sizes. We show the necessity of our modifications and explain how the improvement is due to a probabilistic quadratic speed-up in the convergence of core-set loss, under assumptions about the relationship of model uncertainty and misclassification.

## 1 INTRODUCTION

Active learning aims to identify the most informative data to label and include in supervised training. Often, these algorithms focus on reducing model variance, representing distributional densities, maximizing expected model change, or minimizing expected generalization error (Kirsch et al., 2019; Sener & Savarese, 2018; Settles, 2009; Shen et al., 2018; Sinha et al., 2019). A unifying theme is efficient data collection, which is measured by the rate in improvement as more data are labelled. This is important when we want to identify only the most promising samples to be labelled, but also for tasks that require slow or expensive labelling (Ducoffe & Precioso, 2018; Ma et al., 2020; Settles, 2009).

We describe active learning with the same notation as Sener & Savarese (2018). Suppose we wish to classify elements of a compact space $\mathcal{X}$ into labels $\mathcal{Y} = \{1, \ldots, C\}$. We collect $n$ data points $\{x_i, y_i\}_{i \in [n]} \sim P_{\mathcal{X} \times \mathcal{Y}}$, but only have access to the labels of $m$ of these, denoted by their indices $s = \{s^{(i)} \in [n]\}_{i \in [m]}$. We use the learning algorithm $A_s$ on the labelled set $s$ to return the optimized parameters of the classifier, and measure performance with the loss function $l(\cdot, \cdot; A_s) : \mathcal{X} \times \mathcal{Y} \to \mathbb{R}$. The goal of active learning is to produce a set of indices $s_+$ whose cardinality is limited by the labelling budget $b$, such that expected loss is minimized upon labelling and training on these elements: $\arg\min_{s_+ : |s_+| \leq b} \quad \mathbb{E}_{x, y \sim P_{\mathcal{X} \times \mathcal{Y}}} \left[ l(x, y; A_{s \cup s_+}) \right]$ (Sener & Savarese, 2018). In practice, we use the test set $\{x_i, y_i\}_{i \in [t]} \sim P_{\mathcal{X} \times \mathcal{Y}}$ to approximate the expectation. The typical way to assess the data-efficiency of any particular active learning algorithm is to compare its trend of test performance across increasing labels compared to random and other sampling baselines (Kirsch et al., 2019; Sener & Savarese, 2018; Settles, 2009; Shen et al., 2018; Sinha et al., 2019; Ducoffe & Precioso, 2018; Ma et al., 2020).

Sener & Savarese (2018) suggested that we can improve data-efficiency by minimizing the *core-set radius*, $\delta$, defined as the maximum distance of any unlabelled point from its nearest labelled point: $\delta = \max_{i \in [n]} \min_{j \in s} \delta(x_i, x_j)$. Given the generalization error $\zeta_n$ of all labelled *and* unlabelled data, and zero training error, expected error converges linearly with respect to $\delta$ (Sener & Savarese, 2018):

$$\mathbb{E}_{x, y \sim P_{\mathcal{X} \times \mathcal{Y}}} [l(x, y; A_s)] \leq \zeta_n + \frac{1}{n} \sum_{i \in [n]} l(x_i, y_i) \leq \zeta_n + \mathcal{O}\left(C\delta\right) + \mathcal{O}\left(\sqrt{\frac{1}{n}}\right) \tag{1}$$

Sener & Savarese (2018) argued that generalization error for neural networks has well-defined bounds, so optimizing the rest of Equation 1, referred to as *core-set loss*, is critical for active learning. Indeed, their algorithms for optimizing core-sets consistently improved over their baselines (Sener & Savarese, 2018).

Uncertainty-based sampling is particularly valuable for identifying support vectors, leading to finer classification boundaries (Kirsch et al., 2019; Settles, 2009). However, these methods may catastrophically concentrate their labelling budget on difficult, noisy regions between classes, as shown in Figure 1.

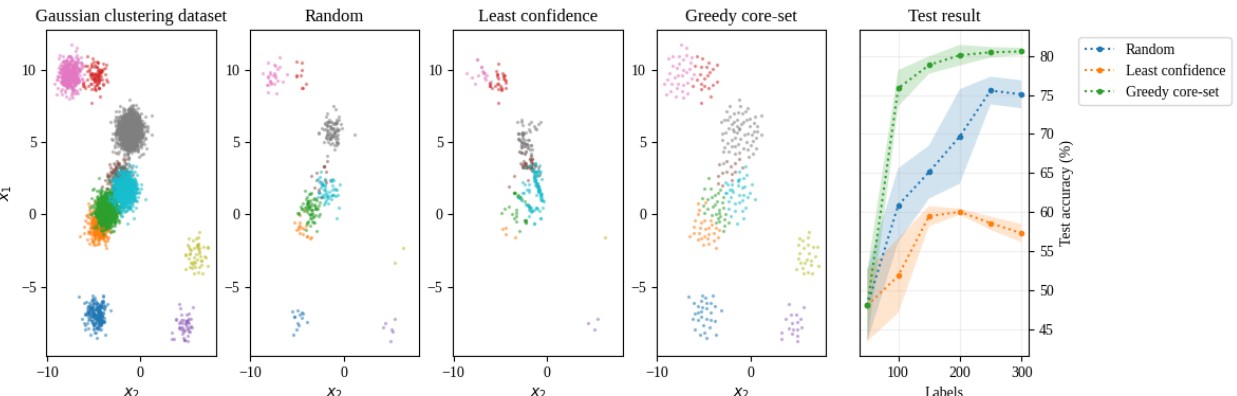

Figure 1: A toy demonstration of catastrophic concentration from least confidence acquisition functions. Columns 2, 3 and 4 show the labelling requests made by three different active learning strategies over 5 iterations on the dataset shown in column 1. Each strategy has a request budget of 50 labels per iteration. Test results show the mean and 1 standard deviation of 3 random initializations. Whereas least confidence wastes its labelling budget on difficult regions between clusters and performs worse than random acquisition, the greedy core-set strategy covers the input space effectively and achieves superior label efficiency.

We present a two-part solution for incorporating uncertainty into core-sets:

1. Scale distances between points by *doubt* ($I$) (Settles, 2009; Shen et al., 2018) before computing core-set radii:

$$\hat{\delta}_i = \delta_i \, I(x_i), \text{ where } I(x) = 1 - \max_y P(y|x) \tag{2}$$

2. Apply beam search to greedily identify the core-set configuration among $K$-candidates with the lowest maximum log-confidence to reduce the variance of core-set trajectories.

## 2 BACKGROUND

**Greedy versus optimal core-set for active learning.** Core-set radius $\delta$ is the maximum of all distances between each data point in $x_u = \{x_i : \forall i \in [n]\}$ and its closest labelled point in $x_l = \{x_i : \forall i \in s\}$ (Sener & Savarese, 2018). The optimal core-set achieves linear convergence of core-set loss in respect to $\delta$ by finding the acquisition set $s_+$ with optimal core-set radii $\delta_{OPT}$ shown in Equation 3 (Sener & Savarese, 2018). Sener and Savarese used $l_2$-norm between activations of the last layer of VGG16 as $\Delta$.

$$\delta_{OPT} = \min_{s_+} \max_{i \in [n]} \min_{j \in s_+ \cup s} \Delta(x_i, x_j) \quad (3) \qquad \delta_g = \max_{i \in [n]} \min_{j \in \hat{s_+} \cup s} \Delta(x_i, x_j) \le 2\,\delta_{OPT} \quad (4)$$

Since this problem is NP-hard (Cook et al., 1998), Sener & Savarese (2018) proposed a greedy version shown in Algorithm 1 with acquisitions $\hat{s_+}$ bounded above by Equation 4. It returns a selection mask over the data pool to signal labelling requests for elements that greedily minimize the maximum distance between any point and its nearest labelled point.

**Algorithm 1:** Greedy core-set (Sener & Savarese, 2018)

```
1  def greedy core-set (xu, xl, budget):
2      selection = [0 : ∀i ∈ [|xu|]];
3      for t = 0, ..., budget do
4          i =
                arg max_{i∈[|xu|]} min_{j∈[|xl|]} Δ(xu^(i), xl^(j));
5          selection^(i) = 1;
6      return selection;
```

Figure 2 shows how $\delta$ varies compared to closest-K-means core-sets, where the core-set consists of the closest points to optimized K-means.

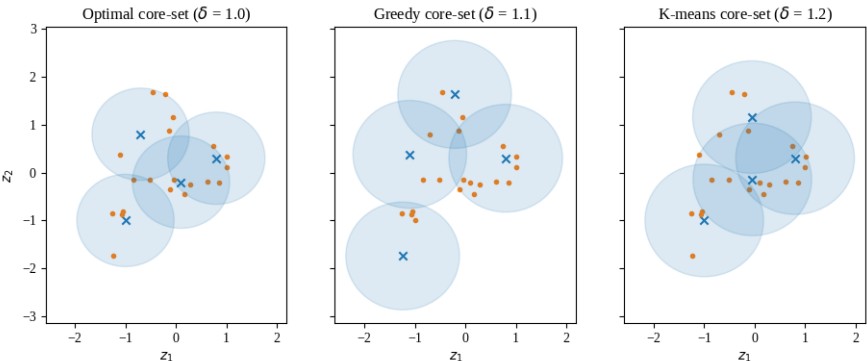

Figure 2: Core-set radii ($\delta$) differs between optimal, greedy and closest-K-means core-sets ($K = 4$). Crosses and dots represent the labelled and unlabelled set, respectively, and translucent circles show the $\delta$-cover. Sener & Savarese (2018) found a $\delta$-linear upper bound on core-set loss convergence.

**Related techniques for batched acquisition.** Batch active learning by diverse gradient embeddings acquires batches in two steps. First, we compute loss gradients in respect to the parameters of the last layer of the classifier for each unlabelled point and its most probable label (Ash et al., 2020). Then, we sample from clusters of these gradients using, for instance, K-means++ to avoid catastrophic concentration (Ash et al., 2020). We share similar intuitions about classifier confidence and intra-batch diversity being important sources of information that may enhance active learning, but differ in that we do not optimize for model change and use core-sets for diversification because of its theoretical foundations.

BatchBALD acquires batches that maximize the mutual information between the joint data and model parameters and was intended to overcome redundant sampling of repeated BALD (Kirsch et al., 2019). We tried combining this with a probabilistic technique of estimating likely core-set locations (see Appendix A.2), but it appeared that core-sets mainly require $\delta$ minimization for core-set loss convergence.

## 3 METHODS

Algorithm 2 and its dependence on Algorithm 3 implement doubt-weighted greedy core-set to run on GPU. To incorporate uncertainty information, we make two key changes to the original greedy core-set algorithm. First, we compute core-sets in a warped space where distances originating from any unlabelled point diminish to zero with classification confidence.

Given inputs $x_u$ and $x_l$, which represent the unlabelled and labelled data, Line 2 of Algorithm 2 calls Algorithm 3 to pre-compute distances of each unlabelled datum to their nearest labelled data. We scale these distances by the doubt of the classifier on the respective unlabelled data on Lines 3 and 11. Each acquisition is removed from the existing unlabelled pool, and Line 12 updates new core-set radii. Figure 3 illustrates how core-sets in these spaces preferentially cover regions of low confidence.

---

**Algorithm 2:** Doubt-weighted core-set

1   **def** `doubted_core-set` ($x_u$, $x_l$, *batch_size, budget*):
2     $\min\_\delta$ = `compute_min_`$\delta$ ($x_u$, $x_l$, *batch_size*);
3     $\min\_\hat{\delta} = [\min\_\delta^{(i)} \cdot I(x_u^{(i)}) : \forall i \in [|x_u|]]$;
4     $x_u$ = `memory-copy` ($x_u$);
5     $index = [i : \forall i \in [|x_u|]]$;
6     $selection = [0 : \forall i \in [|x_u|]]$;
7     **for** $t = 0, \ldots,$ *budget* **do**
8       $i = \arg\max \min\_\hat{\delta}^{(i)}$;
9       $selection^{(index[i])} = 1$;
10      $x = x_u^{(i)}$;
       **splice out:** $index^{(i)}, x_u^{(i)}, \min\_\delta^{(i)}$
11      $\hat{\delta} = \Delta(x, x_u) \cdot I(x)$;
12      $\min\_\hat{\delta} = \left[ \min\{\min\_\hat{\delta}^{(k)}, \hat{\delta}^{(k)}\} : \forall k \in [|x_u|] \right]$
13    **return** $selection$;

---

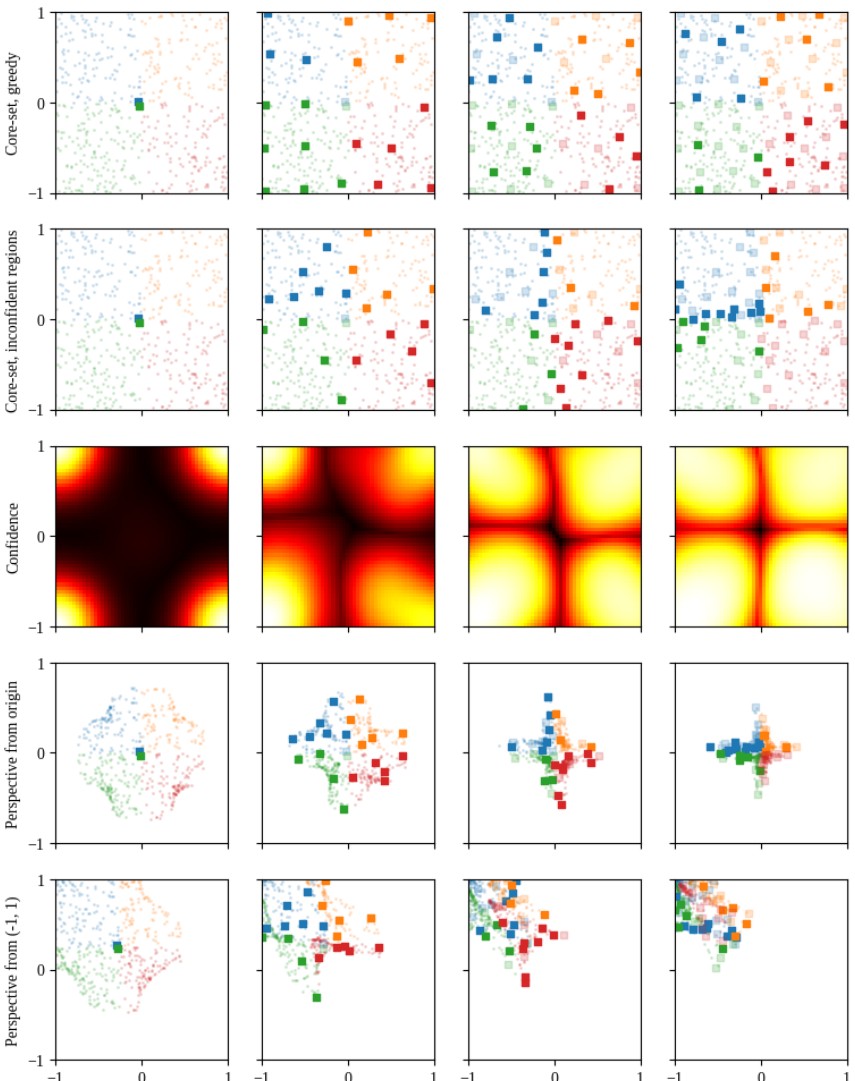

Figure 3: Quadrant classification of $\mathbf{x} \in [-1, 1]^2$, represented as translucent dots. The first column shows the initial dataset and model confidence. Columns progress from left to right at a rate of 20 label acquisitions per step. Solid squares represent current acquisitions, while translucent squares represent previous acquisitions. Row 1 shows that the core-sets maximize uniform coverage of the feature space. In contrast, rows 2 and 3 show preferential coverage of class boundaries when we apply the same algorithm to point-wise distances that are scaled by the doubt of the model. Rows 4 and 5 show how this new method maximizes uniform coverage in the uniquely warped space perceived from candidate points in an uncertain (origin) versus high-confidence (-1, 1) region. Note how the furthest distances from the origin are along the axes (i.e., the furthest distances from (-1, 1) occur along the boundary of the red quadrant with the green and orange quadrants, rather than (1, -1)).

We choose the same $\Delta$ as Sener & Savarese (2018), which is $l_2$-norm between activations of the last layer of VGG16. Given unlabelled data of size $U = |x_u|$, labelled data of size $L = |x_l|$, feature size $\forall i \in [U], \forall j \in [L], D = \dim x_u^{(i)} = \dim x_l^{(j)}$, batch size $B \ll \min\{U, L\}$ and labelling budget $b$, Algorithm 3 costs $\Theta\left(ULD\right)$ steps and $\Theta\left(B^2D\right)$ memory with the bottleneck on line 5. Excluding line 2, Algorithm 2 costs $\mathcal{O}\left(bUD\right)$ in both computation and memory with the bottleneck on line 11. Since both the original core-set algorithm and our modification requires fine-tuning VGG16 per addition to the training set, and computing the class probabilities requires only a single linear transformation, the final computational complexity is the same as the original core-set search. For core-set sizes 5k to 15k, compute time scales linearly from 25 s to 50 s on a NVIDIA Titan GPU.

---

**Algorithm 3:** Memory-efficient core-set radii

---

1 **def** compute_min_$\delta$ ($x_u$, $x_l$, $b$):
2    min_$\delta = [\infty : \forall k \in [|x_u|]]$;
3    **for** $i = 0, \ldots, \frac{|x_u|}{b}$ **do**
4       **for** $j = 0, \ldots, \frac{|x_l|}{b}$ **do**
5          $\tilde{\delta}_{i:i+b,j:j+b} = \Delta(x_u^{(i:i+b)}, x_l^{(j:j+b)})$;
6          min_$\tilde{\delta}_{i:i+b} = \left[\min_{c\in[j,j+b]} \tilde{\delta}_{i:i+b,j:j+b}^{(k,c)} : \forall k \in [i, i+b]\right]$;
7          min_$\delta^{(i:i+b)} = \left[\min\left\{\text{min}\_\delta^{(k)}, \text{min}\_\tilde{\delta}_{i:i+b}^{(k)}\right\} : \forall k \in [i, i+b]\right]$
8    **return** min_$\delta$;

---

Second, we use beam search to greedily prune and keep track of the top resulting core-set configurations with the lowest overall confidence. Since there is no guarantee for the optimality of greedy core-sets (Sener & Savarese, 2018), we seek an orientation with the most points near classification regions of high uncertainty at the cost of increasing compute and memory complexity by a factor of the beam width.

We modify maximum normalized log probability (Shen et al., 2018) to rank overall classifier uncertainty $\mathbb{U}$ of core-set $s$:

$$\mathbb{U}(s) = -\frac{1}{|s|}\sum_{x\in s}\log(1 - I(x)) \quad (5)$$

Figure 4 shows a sample ranking of the configurations found during beam search with width $K = 4$.

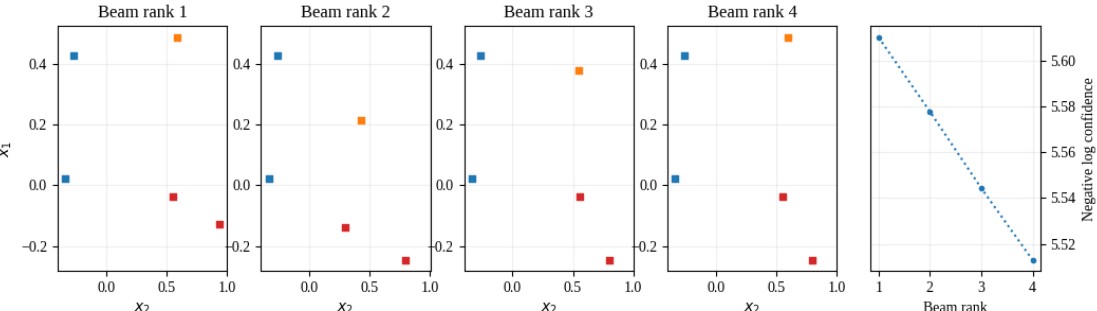

Figure 4: A sample acquisition from the toy quadrant dataset in Figure 3. Beam search for the top greedy core-sets ranks the configurations by increasing overall confidence (i.e., decreasing negative log confidence). Regions of high uncertainty occur near the $x_1 = 0$ and $x_2 = 0$ axes in this dataset. The left-most configuration (beam rank 1) should be selected for its dual optimality of core-set radii and low overall confidence.

**Active learning pipeline.** For each active learning experiment, we start by randomly partitioning the full training set into an initial pool of labelled data and an unlabelled pool of features. We fine-tune the parameters of a ImageNet-1-pretrained VGG16 on this initial dataset. For each training batch size of 64, we optimize for cross-entropy loss using Adam (Kingma & Ba, 2015) under default hyperparameters from PyTorch (Paszke et al., 2019) and a learning rate of 0.01 for CIFAR10/100 and 0.005 for SVHN. We then use either random acquisition, the original greedy core-set algorithm, or variations of Algorithm 2 with the trained model to produce a selection mask over the unlabelled data. We enforce that the number of selected elements is equal to the labelling budget per iteration. The selected features and their labels join the training set, the model retrains with a re-initialized optimizer, and the process is repeated until the number of the labelled data reaches the specified ceiling for the experiment.

Note that we do not compare with the other baselines used in the original core-set experiments. Since the original core-set algorithm improved significantly from those baselines, we expect improvement over the original core-set algorithm to imply similar or greater improvement as well.

Table 1 shows the iterations that we found were necessary to roughly meet zero training error on the initial dataset ("First-pass") and all additions to the dataset per active learning iteration ("Thereafter"). Note that validation error is not required to satisfy the convergence requirement of core-sets, so we ignore it in our experiments.

Table 1: Epochs of optimization required to consistently surpass 99% training accuracy.

| Dataset | Training epochs | |
| --- | --- | --- |
| | First-pass | Thereafter |
| CIFAR10 | 30 | 12 |
| CIFAR100 | 80 | 20 |
| SVHN | 50 | 20 |

For the ablation studies, we tune hyperparameters and conduct ablation studies on CIFAR10 (Krizhevsky, 2009) using a budget of 400 labels per active learning iteration and an initial dataset size of 1000 samples. We use the same hyperparameters as the ablation studies in the main experiments on CIFAR10/100 (Krizhevsky, 2009) and SVHN (Netzer et al., 2011), which uses a budget of 5000 labels per iteration and a starting dataset size of 5000 samples.

## 4 RESULTS

Figure 5 shows the results of ablation studies on the small-scale version of the main experiments, where we observe that beam search for the core-set configuration with the lowest log confidence yields significant improvements over greedy core-set only if core-set radii are scaled by the uncertainty of each unlabelled sample.

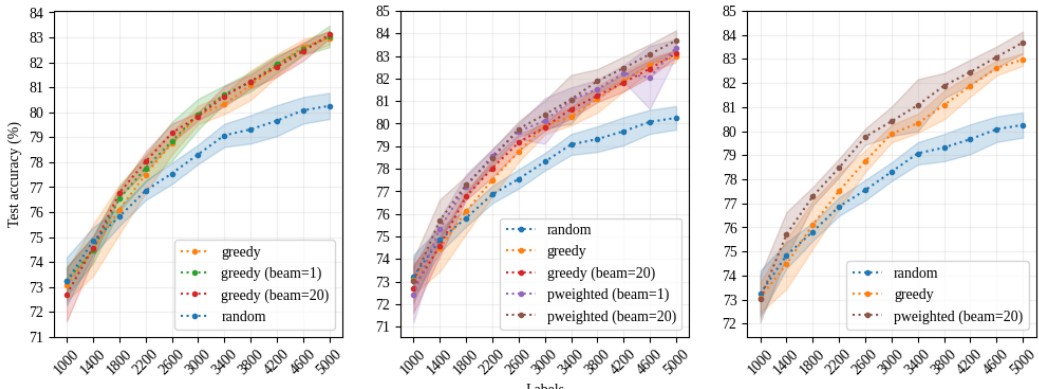

Figure 5: Ablation results on CIFAR10 showing the mean and 1 standard deviation derived from 5 random initializations. *Left:* using beam search (beam=20) to greedily find the core-set configuration with the lowest log confidence has no effect. *Middle:* weighing perceived distances by uncertainty (pweighted) results in better performance and beam search reduces variance, resulting in the highest final scores. Note that using a single beam in "pweighted" resulted in occasionally deteriorating performance, which was avoided using 20 beams. *Right:* performance improvement over the original greedy core-set algorithm appears significant.

Figure 6 shows how our contributions significantly improve the label efficiency of greedy core-set on CIFAR10 and SVHN on large-scale active learning experiments under the same hyperparameters from the ablation studies. Our contributions increase absolute label efficiency above random acquisition.

**Theoretical rationale for improved label efficiency using confidence-weighted distances.** Sener & Savarese (2018) showed that the softmax function over $c$ classes is Lipschitz continuous and we denote its constant as $\lambda_c$. We define confidence to be the max of the softmax output and assume that the confidence of any training point is 1. Consider an unspecified unlabelled point $x_u$ that is located $r$ distance away from its closest labelled point $x_i$ in the training set $s$. Equation 6 shows the bound on doubt $I$ (i.e., 1 minus confidence) as a function of distance $r$ from the nearest training example. We interpret this to be rising minimum uncertainty with increasing distance from the nearest training example, capped at 1.

$$I(r) \leq \min\{1, r\lambda_c\} \text{, where } r = \min_{i \in s} \Delta(x_i, x_u) \tag{6}$$

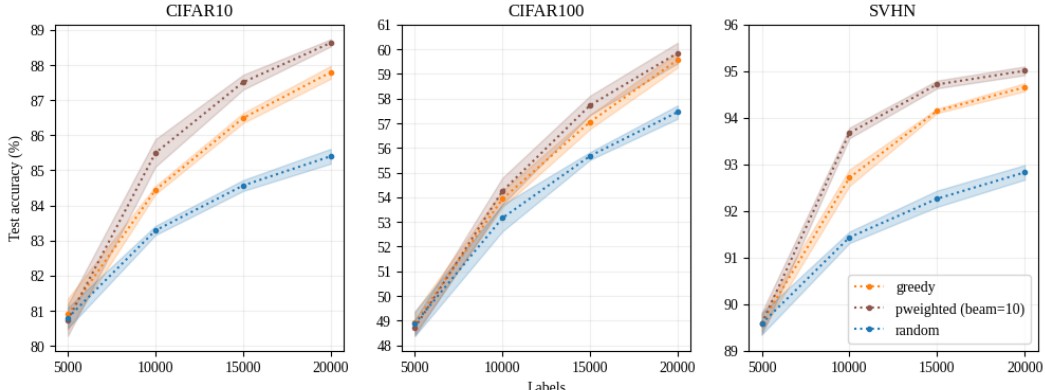

Figure 6: Means and 1 standard deviation derived from 5 random initializations on image classification using the same acquisition budgets as Sener & Savarese (2018). Weighing perceived distances by uncertainty (pweighted) and using beam search (beam=10) to find the core-set configuration with the lowest log confidence causes significant active learning improvement over vanilla core-sets.

Recall that we scale $\delta$ by doubt to obtain a new radius, $\hat{\delta}$. Originally, $\delta$ was the minimum distance between each unlabelled point and its nearest labelled point; **in our case, we define $\hat{\delta}$ to be the minimum distance between each unlabelled point and its nearest *unlabelled point with 0 error* that we know exists with probability $\beta$.** Then, Equation 7 holds with probability $\beta$.

$$\hat{\delta} = \delta I(\delta) \leq \begin{cases} \delta & \text{if } \delta\lambda_c \geq 1 \\ \delta^2\lambda_c & \text{if } \delta\lambda_c < 1 \end{cases} \tag{7}$$

To clarify, we assume that for any unlabelled point, with probability $\beta$ there exists another nearby unlabelled point that behaves as if it exists in the training set already. Instead of using the distance from the nearest labelled point, the core-set loss can use the distance from this unlabelled point instead. Equation 8 shows that with probability $\beta$, the convergence of the core-set loss for all datasets where $\delta\lambda_c < 1$ now depends on a quadratic factor of $\delta$ versus the linear relationship from before.

$$\frac{1}{n}\sum_{i\in[n]} l(x_i, y_i; A_s) \leq \mathcal{O}\left(C\hat{\delta}\right) + \mathcal{O}\left(\sqrt{\frac{1}{n}}\right) = \mathcal{O}\left(C\delta^2\lambda_c\right) + \mathcal{O}\left(\sqrt{\frac{1}{n}}\right), \text{ only if } \delta\lambda_c < 1 \quad (8)$$

Recall that the greedy core-set algorithm minimizes $\delta$ towards 0 per optimization step. This means that larger initial training sets should benefit our algorithm more, since the smaller $\delta$ will more likely yield a quadratic convergence of core-set loss.

Next, we analyze the probability $\beta$ of such an unlabelled point with 0 error existing in a radius $\hat{\delta}$ around each unlabelled point. In order to do this, we make 2 key assumptions about the relationship between model confidence and empirical misclassification. Assume that the probability of incorrect classification is equal to the product of doubt with $\epsilon$, which represents the error rate given doubt. We further assume that $\epsilon$ equals 0 starting at any training point and is $\lambda_\epsilon$-Lipschitz continuous for any distance extending away the closest training point, as shown in Equation 9. To deduce the probability $\beta$ of at least one unlabelled point with 0 error existing between the given unlabelled point and a distance $r_0$ from its nearest labelled point, we subtract from 1 the probability of non-zero error occurring in all these unlabelled points, which Equation 10 bounds (see proof in Appendix A.1, Claim 1).

$$P_{\text{err}}(r) = I(r) \cdot \epsilon(r) \leq \lambda_c\lambda_\epsilon r^2 \quad (9) \qquad \beta = 1 - \prod_{r_0}^{\delta} P_{\text{err}}(r)^{dr} \geq 1 - \frac{(\lambda_c\lambda_\epsilon)^{\delta-r_0}}{\exp(\delta-r_0)^2}\left(\frac{\delta^\delta}{r_0^{r_0}}\right)^2 \quad (10)$$

Now suppose $r_0 = \delta z$, where $z \in [0, 1]$ is a scaling factor of $\delta$. Then:

$$\beta \geq 1 - \left(\left(\frac{\delta\sqrt{\lambda_c\lambda_\epsilon}}{e}\right)^{1-z}\frac{1}{z^z}\right)^{2\delta} \qquad \text{(see Appendix A.1: Claim 4)} \qquad (11)$$

We will have no information on whether our algorithm improves upon vanilla core-sets when $\beta \geq 0$, or when we rely on random chance that there exists an unlabelled point with 0 error within $\hat{\delta}$ distance from any unlabelled point of interest. Equation 12 shows the minimum distance $\delta^*$ such a point would have to be from the nearest labelled point:

$$\beta \geq 0 = 1 - \left( \left( \frac{\delta^* \sqrt{\lambda_c \lambda_\epsilon}}{e} \right)^{1-z} \frac{1}{z^z} \right)^{2\delta} \quad \longrightarrow \quad \delta^* \geq \frac{e \, z^{\frac{z}{1-z}}}{\sqrt{\lambda_c \lambda_\epsilon}} \tag{12}$$

Figure 7 shows slices of the lower bound on $\beta$ from Equation 11. When confidence is high (i.e., low $z$) for an unspecified unlabelled point located $\delta$ from its closest labelled point, we expect higher probabilities for an unlabelled point with 0 error to exist between $\delta z$ and $\delta$. The figure also illustrates an increase in decay rate of this probability with decreasing confidence. This makes sense because we assumed that confidence, to some degree, indicates correctness of the model on unlabelled data.

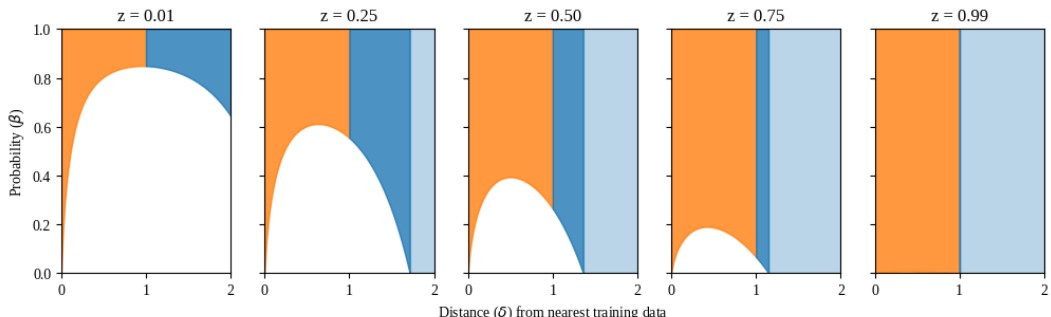

Figure 7: We fix $\lambda_c = \lambda_\epsilon = 1$ and observe the probability $\beta$ of our algorithm performing better than the original greedy core-set algorithm across a range of distances $\delta$ from the nearest training point at different slices of doubt $z$. All colored regions represent possible values of $\beta$. Orange regions occur when $\delta < 1/\lambda_c$, which represent the probability of quadratic convergence of core-set loss in respect to $\delta$ (see Equation 8). All blue regions indicate values of $\beta$ in which our algorithm performs at least as well as greedy core-set search. Pale blue regions occur when $\delta > \delta^*$ (see Equation 12), which represent areas where we cannot reason about $\beta$.

There is less information to extract about the surroundings of points with low confidence, so the lower bound on $\beta$ naturally flattens sooner, resulting in shorter $\delta^*$ and a larger space in which we cannot infer any benefit of our algorithm over vanilla greedy core-sets. The decay of the lower bound close to the origin was also expected, since the space between $\delta z$ and $\delta$ rapidly diminishes into 0 when $\delta$ shrinks, which does not allow much opportunity for an unlabelled point with 0 error to appear.

## 5 DISCUSSION

Sener & Savarese (2018) suspected the potential of incorporating uncertainty information to improve core-sets for active learning and we successfully confirm this in our implementation of greedy core-set search on doubt-scaled distances. Our ablation studies show that doubt-scaling is critical for fast core-set loss convergence while beam search for the core-sets with low overall confidence stabilizes acquisition variance. Assuming that doubt acts as a cheap but noisy estimate of the distance to the nearest point with zero error, the theoretical results show that our empirical improvements is caused by a probabilistic quadratic-minimization of $\delta$ improving the linear order from before.

The difference between core-set loss convergence of the ablation studies versus full experiments on CIFAR10/100 and SVHN is explained by differences in their dataset and budget sizes. Larger core-sets, whose quality consists of the diversity of the initial, uniformly-sampled dataset and future acquisitions, have smaller $\delta$ because the maximal distance between any unlabelled point and its nearest labelled point is naturally minimized as the goal of any core-set algorithm. Since $\delta$-quadratic convergence of core-set error can only occur when $\delta$ is sufficiently small (i.e., $\delta < 1/\lambda_c$), it is expected that our contributions improved the difference in core-set loss using the larger core-sets of the full experiments. The smaller initial datasets and acquisitions of the ablation studies would have

large $\delta$ that may exceed the threshold-criteria for $\delta$-quadratic convergence or even $\delta^*$, for which we will have no guarantee of improvement above vanilla greedy core-set search.

Our theoretical results also explain why the rates of performance improvement from our method appear to diminish faster with more data. When the labelled set saturates its coverage of the full distribution, $\delta$ diminishes towards 0. The lower bound on the probability of $\delta$-quadratic convergence diminishes to 0 regardless of model confidence in this region (see Figure 7), preventing us from reasoning about the benefit of our algorithm. Intuitively, when the training set is sufficiently large and varied such that it already covers the vast majority of the input distribution, confidence estimations may be too similar to distinguish a signal about $\delta$ from noise.

## 6 CONCLUSION

Greedy core-set search in doubt-scaled space empirically and theoretically improves upon the original algorithm in active learning. We show that the magnitude of improvement is greatest for datasets that are not too small or already comprehensive of the input distribution, which maximizes the probability of quadratic convergence of core-set loss with respect to core-set radii minimization. Even in cases where the performance of our contribution equates to that of the original core-set algorithm, there is no additional computational cost.

The value of our algorithm is a strict improvement over the original core-set, such that active learning performance improves with more labels. We suspect that our algorithm would most benefit online, large-scale active learning systems.

## 7 REPRODUCIBILITY STATEMENT

See supplementary code to replicate all results.

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

## A  APPENDIX

### A.1  ROUGH WORK FOR CLAIMS

**Claim 1.**

$$\beta \geq 1 - \frac{(\lambda_c \lambda_\epsilon)^{\delta - r_0}}{\exp(\delta - r_0)^2} \left( \frac{\delta^\delta}{r_0^{r_0}} \right)^2 \tag{13}$$

*Proof.*

$$\beta = 1 - \prod_{r_0}^{\delta} P_{\text{error}}(r)^{dr} = 1 - \exp\left( \int_{r_0}^{\delta} \ln P_{\text{error}}(r) dr \right) \tag{14}$$

$$\geq 1 - \exp\left( \int_{r_0}^{\delta} \ln \lambda_c \lambda_\epsilon r^2 dr \right) \qquad \text{Lower bound from Eq. 9} \tag{15}$$

$$= 1 - \exp\left( r \ln(\lambda_c \lambda_\epsilon r^2) - 2r \Big|_{r_0}^{\delta} \right) \qquad \text{(see Appendix A.1: Claim 2)} \tag{16}$$

$$= 1 - \frac{(\lambda_c \lambda_\epsilon)^{\delta - r_0}}{\exp(\delta - r_0)^2} \left( \frac{\delta^\delta}{r_0^{r_0}} \right)^2 \qquad \text{(see Appendix A.1: Claim 3)} \tag{17}$$

$\square$

**Claim 2.**

$$\int \ln(ax^2) \, dx = x \ln(ax^2) - 2x + C$$

*Proof.* Use integration by parts. Let:

$$f(x) = ax^2 \qquad u = \ln f(x) \qquad dv = dx$$
$$f'(x) = 2ax \qquad du = f'(x)/f(x)dx \qquad v = x$$

Also note:

$$\frac{x f'(x)}{f(x)} = \frac{2ax^2}{ax^2} = 2$$

So the original problem can be integrated by parts:

$$\int \ln(ax^2) dx = \int u \, dv$$

$$= uv - \int v \, du$$

$$= x \ln f(x) - \int \frac{x f'(x)}{f(x)} dx$$

$$= x \ln(ax^2) - 2x + C$$

$\square$

**Claim 3.**

$$\exp\left( r \ln(\lambda_c \lambda_\epsilon r^2) - 2r \Big|_{r_0}^{\delta} \right) = \frac{(\lambda_c \lambda_\epsilon)^{\delta - r_0}}{\exp(\delta - r_0)^2} \left( \frac{\delta^\delta}{r_0^{r_0}} \right)^2$$

*Proof.*

$$\exp\left(r\ln(\lambda_c\lambda_\epsilon r^2) - 2r\Big|_{r_0}^{\delta}\right) = \exp\left(\delta\ln(\lambda_c\lambda_\epsilon\delta^2) - r_0\ln(\lambda_c\lambda_\epsilon r_0^2) - 2\delta + 2r_0\right)$$

$$= (\lambda_c\lambda_\epsilon\delta^2)^\delta(\lambda_c\lambda_\epsilon r_0^2)^{-r_0}\exp\left(-2(\delta - r_0)\right)$$

$$= \frac{(\lambda_c\lambda_\epsilon)^\delta}{(\lambda_c\lambda_\epsilon)^{r_0}}\frac{\delta^{2\delta}}{r_0^{2r_0}}\frac{1}{\exp\left(\delta - r_0\right)^2}$$

$$= \frac{(\lambda_c\lambda_\epsilon)^{\delta-r_0}}{\exp(\delta - r_0)^2}\left(\frac{\delta^\delta}{r_0^{r_0}}\right)^2$$

$\square$

**Claim 4.**

$$\frac{(\lambda_c\lambda_\epsilon)^{\delta(1-z)}}{\exp(1-z)^{2\delta}}\left(\frac{\delta^\delta}{\delta^{\delta z}z^{\delta z}}\right)^2 = \left(\left(\frac{\delta\sqrt{\lambda_c\lambda_\epsilon}}{e}\right)^{1-z}\frac{1}{z^z}\right)^{2\delta}$$

*Proof.*

$$\frac{(\lambda_c\lambda_\epsilon)^{\delta(1-z)}}{\exp(1-z)^{2\delta}}\left(\frac{\delta^\delta}{\delta^{\delta z}z^{\delta z}}\right)^2 = \left(\frac{\sqrt{\lambda_c\lambda_\epsilon}}{e}\right)^{2\delta(1-z)}\left(\frac{\delta^{\delta(1-z)}}{z^{\delta z}}\right)^2$$

$$= \left(\frac{\delta\sqrt{\lambda_c\lambda_\epsilon}}{e}\right)^{2\delta(1-z)}\frac{1}{z^{2\delta z}}$$

$$= \left(\left(\frac{\delta\sqrt{\lambda_c\lambda_\epsilon}}{e}\right)^{1-z}\frac{1}{z^z}\right)^{2\delta}$$

$\square$

## A.2 Negative result: probabilistic core-sets

Instead of using a deterministic algorithm to compute core-sets, we score random batches on their likelihood of being a subset of the optimal core-set. The goal is for the concatenation of the best-scoring batches with the training set to result in a set of elements that are spread out on the feature space and occur in dense regions, minimizing $\delta$ by definition.

Suppose we are interested in classifying whether a real number is positive or not. Figure 8 shows the unlabelled data and the result of learning a Gaussian mixture model (GMM) over the features. We then use the trained GMM to estimate feature probabilities, which are required for computing modified batch-BALD scores (see Appendix A.3). Figure 9 shows that higher scores indicate features that are likely spread apart. For random batches sampled from this toy dataset, Figure 10 shows that the distribution of scores form a long right tail that contains the most likely core-set centers.

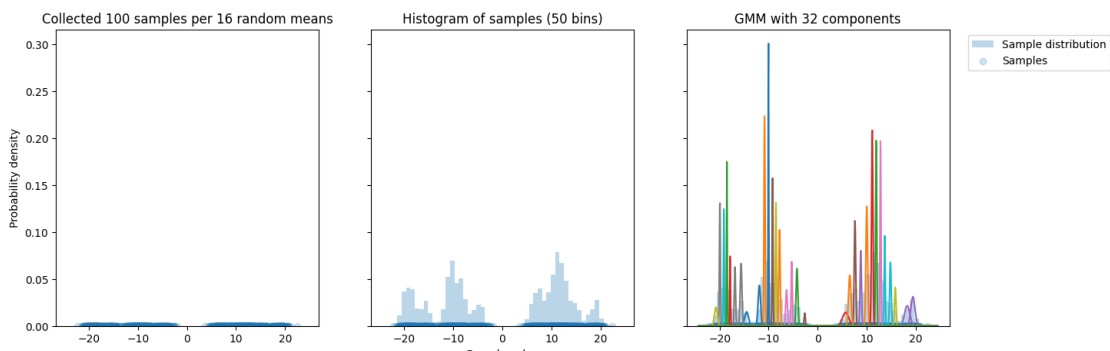

Figure 8: *Left:* collected data with one feature being the value along the real axis. *Middle:* the true distribution of the input features. *Right:* Gaussian mixture with 32 components fit to the collected data. We purposefully overfit the GMM because the distribution of features is not known a priori and we would like high resolution for computing joint information later.

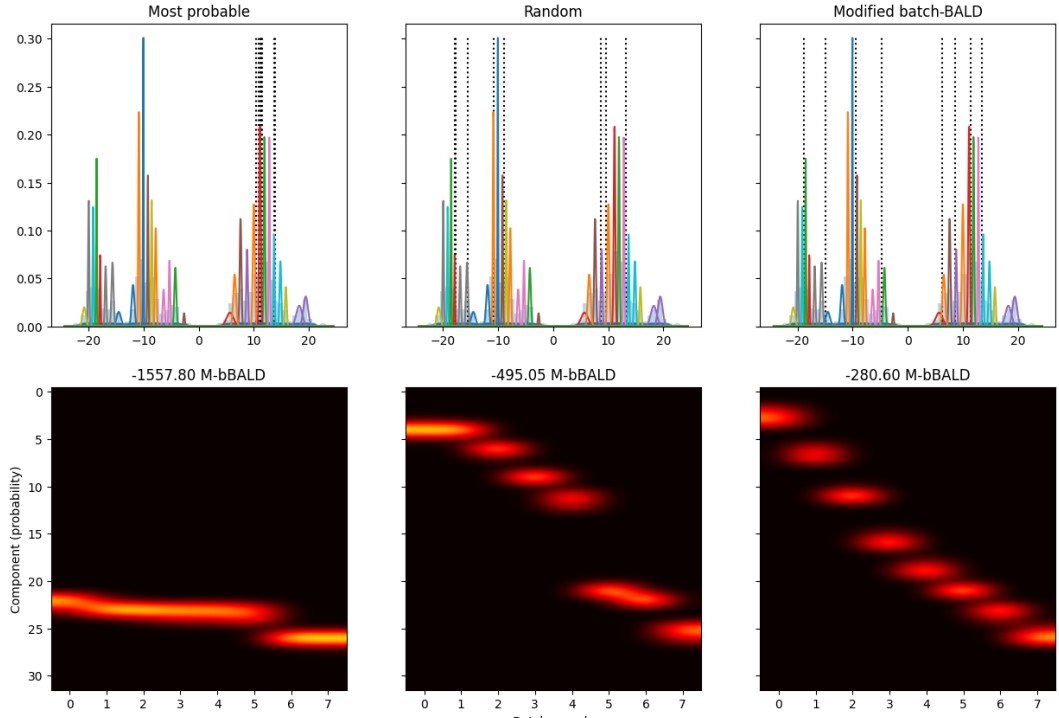

Figure 9: Given the trained Gaussian mixture model from Figure 8, we estimate the probability per component for each element. We use the probabilities to compute a modified batch-BALD joint information score (M-bBALD), which is positively correlated with entropy across the components. The batches with the highest scores occur at dense regions but are spread out across the feature distribution. We want to avoid redundant labelling of elements in the left column. *Top row:* each figure contains eight dotted lines that represent the locations of elements in three different batches. *Bottom row:* corresponding probabilities per GMM component for each element.

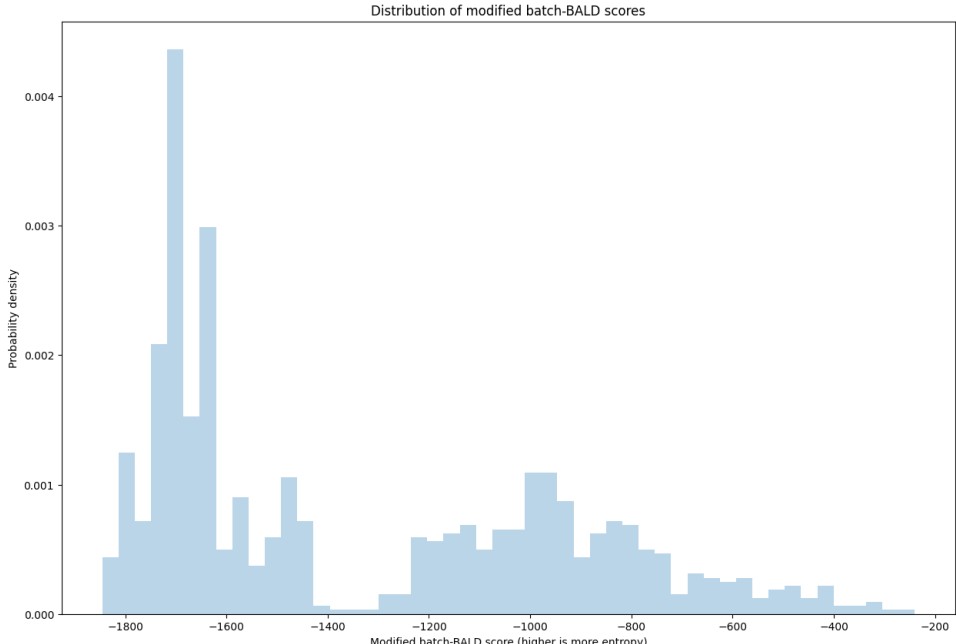

Figure 10: The distribution of modified batch-BALD scores for randomly sampled batches have a long right tail from which we mine for likely core-set elements. Range of scores depends on the number of components in the Gaussian mixture model.

In the subsequent iterations, we first construct a new unlabelled data pool that contains features that have low probability to have appeared in the labelled pool, according to a fitted GMM on the labelled pool. Then, we fit a new GMM on this modified unlabelled pool and repeat the selection algorithm to search for the batch with the highest modified batch-BALD score when combined with the existing training data. We also experiment with interpolating the modified batch-BALD score with the least confidence acquisition metric.

We plot test accuracy versus number of labelled points for random acquisition (random), maximum entropy (max-entropy), least certainty (min-max-probs), probabilistic core-set (probabilistic-coreset) and an exploitative version of probabilistic core-set that interpolates with least certainty at a 9:1 ratio (probabilistic-coreset-exploitive-0.1). Figures 11 and 12 show that there is modest improvement of the core-set variants from the random baseline in both toy datasets, although its significance is unknown. The entropy and least certainty methods performed poorly.

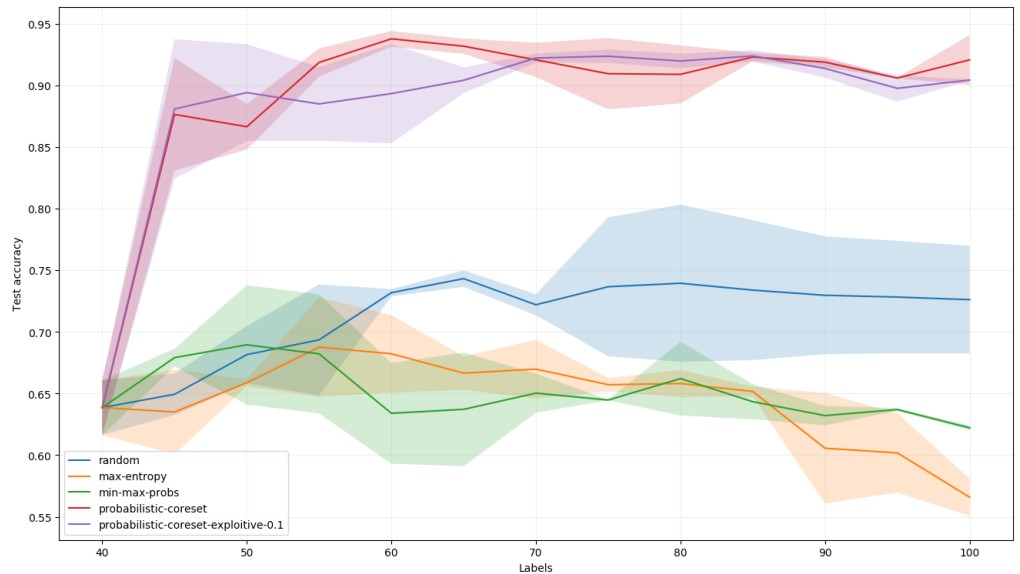

Figure 11: In the toy experiment with 0.5 standard deviation, clusters were mostly separable and core-set variants dominated all baselines. Shaded area represents one standard deviation.

Figure 13 shows examples of data points acquired in the toy experiments by probabilistic core-set versus the points evaluated to be informative by maximum entropy (Figure 14) and least confidence (Figure 15). Whereas the core-set variants prioritized covering the input space, the uncertainty-based methods focused on areas of overlapping clusters, which are prone to error and hard to classify.

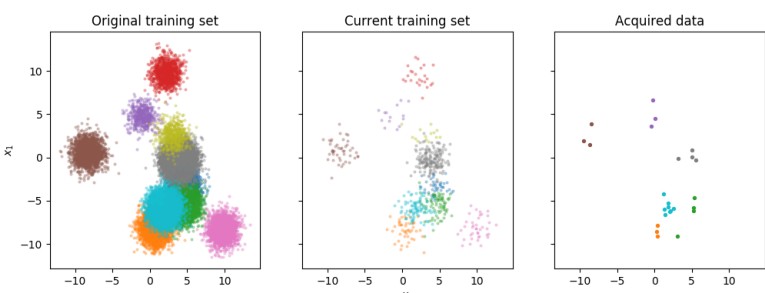

Figure 13: Batch-BALD effectively maximized the distance between elements of selected batches.

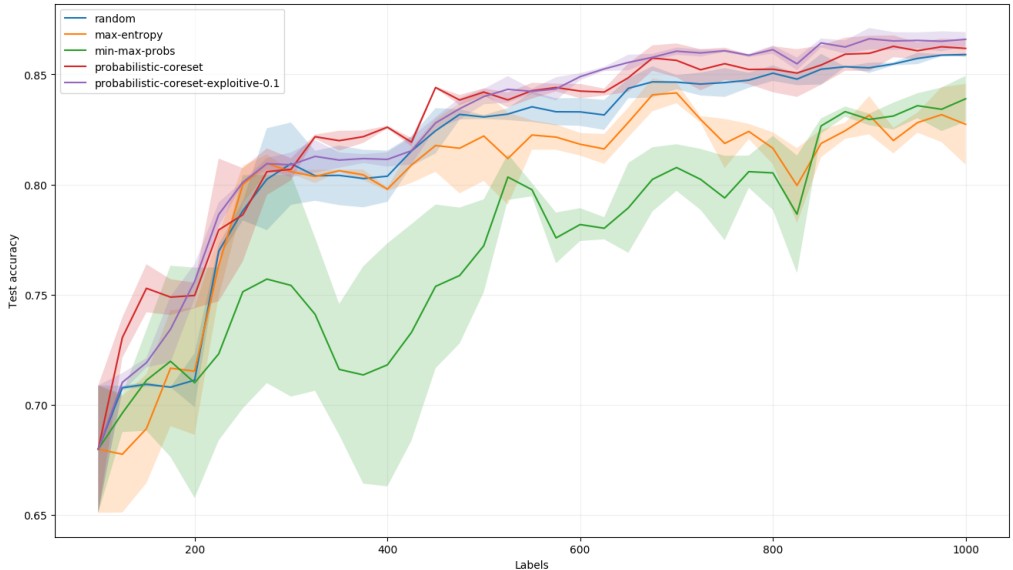

Figure 12: In the toy experiment with 1 standard deviation, clusters overlapped substantially and probabilistic core-set methods formed a modest upper bound in accuracy over all baselines. Shaded area represents one standard deviation.

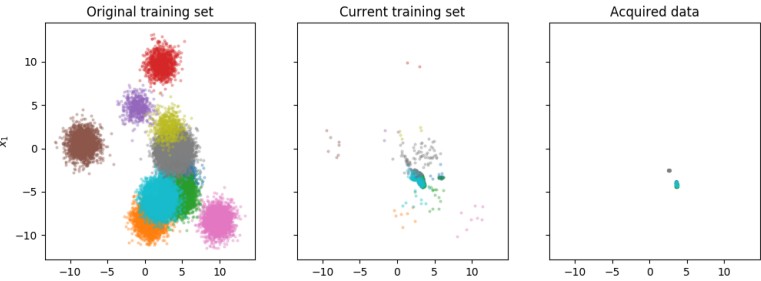

Figure 14: Maximizing entropy resulted in concentrated sampling in the most uncertain regions.

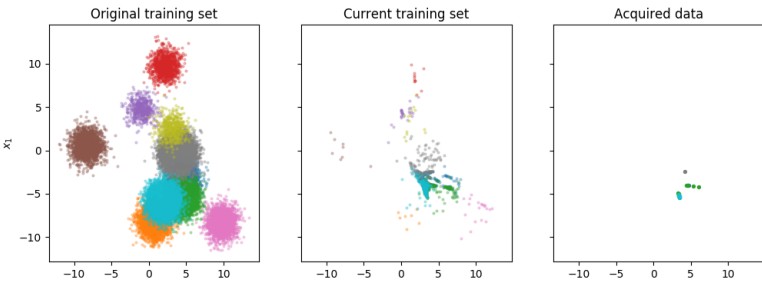

Figure 15: Least confidence also concentrates sampling along uncertain regions.

The poor performance of entropy and uncertainty methods for large batch acquisition in a noisy classification dataset agrees with existing work Settles (2009); Kirsch et al. (2019); Sener & Savarese (2018). The cause of this is wasteful labelling requests in uncertain regions of features that turned out to be inseparable. In contrast, core-set variants and random acquisition are successful because they covered the majority of the input space.

The effect of increasingly difficult separability on acquisition function efficiency is clear in the toy data with 0.5 versus 1 standard deviation. When multiple class distributions overlap substantially, their joint distribution density is sampled more frequently under the core-set variants, which is harm-

ful because those samples do not improve test accuracy for noisy class boundaries. This suggests that class inseparability may play some role in the poor performance of the core-set variants.

Overall, probabilistic core-sets barely improved from random acquisitions and cost more computation than Algorithm 2. Like Sener and Savarese Sener & Savarese (2018), we also conclude with the belief that any method that depends on distributional density sampling will have difficulty exceeding random sampling at an unknown test because of the obvious fact that i.i.d. samples are already well-represented in the target distribution. Then, the main beneficial effect of these density sampling techniques is to reduce redundancy, but this may be a rare phenomenon in the typical high dimensional representations of under-determined and nonlinear classification tasks.

## A.3 BATCH-BALD EVALUATES THE MUTUAL INFORMATION OF BATCHES OF DATA

Given a distribution of model parameters, Bayesian active learning by disagreement (BALD) evaluates the information of a single data point as its marginal entropy penalized with the average entropy across the parameter distribution Kirsch et al. (2019). Intuitively, this selects for samples that elicit low overall certainty from the Bayesian model, but high individual certainty from the competing hypotheses sampled from its parameter distribution. Naive application of BALD to a batch of data may lead to the overestimation of mutual information between elements within the batch Kirsch et al. (2019). On the other hand, Batch-BALD scores their joint information Kirsch et al. (2019).

The Batch-BALD information metric is useful for identifying likely and different core-set centers in two important but different ways from its original setting. First, we fit a GMM and sample its means $\theta$ from $P(\theta)$, which we assume to be uniform. We use these Gaussian means to estimate $P(y|\mathbf{x}, \theta)$. Second, since there may exist multiple means that cover the same peak, optimizing for batch BALD identifies peaks with high overall certainty that have low likelihood of intersecting with other peaks.

