# OpenReview forum: "Improving greedy core-set configurations for active learning with uncertainty-scaled distances"
_ICLR.cc/2022/Conference — ICLR 2022 Submitted_

### Official Review · Reviewer_4LSV · 2021-10-31

**Correctness:** 3
**Technical Novelty And Significance:** 3
**Empirical Novelty And Significance:** 2
**Recommendation:** 3
**Confidence:** 4

**Main Review:**

- The paper is not very well structured. It is hard for me to follow what authors want to say, and notations appear without definition.
- The motivation of adding a probabilistic extension to the existing core set is not well addressed, at least I could not find it.
- As 'memory-efficient' and 'run on GPU' is stated in the paper, is computation efficiency a part of motivation?
- Can the main idea of the 'Theoretical rationale' part be summarized in several sentences? What is more favorable than the original method?
- Moreover, empirical comparison with enough recent methods is lacking. There are methods published after the Sener and Savares paper.

**Summary Of The Paper:**

This paper proposes to improve the core set approach to active learning of Sener and Savares by a probabilistic extension.

**Summary Of The Review:**

This paper needs major revision for judgement.

---

> ### Author Response · Authors · 2021-11-23
> **Some clarifications and answers for your concerns and questions.**
>
> We define all the notations used in the paper in the introduction.
>
> The motivation behind adding a probabilistic extension to core-sets was first suggested as follow up research by Sener and Savarese in their original paper (2018). In our introduction and Figure 3, we also suggest some motivation behind complementing confidence-based approaches with core-sets rather than choosing one extreme over the other.
>
> Figure 7 and the surrounding paragraphs provide a summary of our theoretical section, by describing some scenarios where our method has high probability of out-performing greedy core-sets. In short, there is high probability of quadratic improvement when (a) confidences from the model accurately reflect its probability of error and confidences are moderately high, and (b) training features are adequately diverse such that most unseen data lie within a small radius around their nearest training data.
>
> Our original inquiry and experimental designs intended only to show significant improvement over greedy core-sets at no additional cost in computation complexity. Although the differences are small for smaller acquisition batches (as those seen in Figure 5), the benefit of core-set optimization becomes significant with the larger batches in Figure 6.

---

### Official Review · Reviewer_FMtn · 2021-11-03

**Correctness:** 3
**Technical Novelty And Significance:** 2
**Empirical Novelty And Significance:** 2
**Recommendation:** 5
**Confidence:** 4

**Main Review:**

Pros:
-- Paper well written and clear.
-- Both intuitive and theoretical justifications for why uncertainty-scaled distances could improve upon non-scaled distances.

Cons:
-- The addition of uncertainty based scaling, while interesting is relatively minor. It is not well justified either with respect to results generated.
-- Experimental results remain very unconvincing, with respect to the improvement compared to normal core-sets. And generally experiments are weak. In almost all figures, the performance with scaling is almost equivalent to greedy core-sets (figure 5, and figure 6 on CIFAR100). Seems like for more complicated datasets (CIFAR100) the already small utility of the method vanishes?
-- Non standard active learning set-up where initialisation is VGG16 ImageNet weights rather than randomly initialised weights? Would be good to see performance in this setting.
-- No comparison to other active learning methods. There is significant room for improvement here.
-- Computational cost of these additional steps were analysed.
-- Claim in conclusion "suspect algorithm would most benefit online, large-scale active learning experiment." Results and previous statements just above the conclusion suggest otherwise (this method only assists in very small data regimes over greedy core sets).
-- Why does performance relative to greedy degrade on CIFAR100 (i.e. more difficult task) as compared on CIFAR10 and SVHN? Not much analysis here on that. I suspect something around the usefulness of model's uncertainty with more classes. Some analysis re interaction between uncertainty/model calibration etc. would be useful?

Other questions:
- Clarity: does model uncertainty come from new model or fine-tuned VGG16?
- If from new model (I assumed from finetuned VGG16):, if unoptimised perhaps greedy core-set has the same computational complexity as this method; but greedy k does not for example need to have model trained from previous attempt to calculate distances (and points to choose). The method does increase training time overhead as compared to greedy core-sets?
- If using VGG16 for uncertainty: why? and why not the new model?

**Summary Of The Paper:**

This paper attempts to improve upon the greedy core-set for active learning (Sener and Savarese) by employing distances scaled by uncertainty. The proposed method then leverages a beam search algorithm to identify the best core-set configuration among candidates with the lowest log-confidence to yield further improvements. Empirically evaluated on CIFAR10/100 and SVHN.

**Summary Of The Review:**

Well written paper. However, method is incremental, which is not necessarily bad, but paper does not assist with a weak experimental section and analysis of results.

---

> ### Author Response · Authors · 2021-11-23
> **Answers to your questions and a few clarifications**
>
> Our original inquiry and experimental designs intended only to show significant improvement over greedy core-sets at no additional cost in computation complexity. Although the differences are small for smaller acquisition batches (as those seen in Figure 5), the benefit of core-set optimization becomes significant with the larger batches in Figure 6.
>
> Why does the utility of the new method diminish with CIFAR100 compared to CIRAR10 or SVHN?
> The original core-set loss is bounded by a linear factor of the number of classes (see equation 1, where the notation is defined in paragraph 2 of the introduction), and our method is not immune to this.
>
> Why did we use a non-standard active learning setup where initialization is from VGG-16 weights pre-trained from ImageNet?
> Our main reason was to reduce training time by over 4 fold compared from training from randomly initialized weights. A secondary reason was we believed that the pretrained model would have confidence estimations that better reflected our theoretical assumptions, although this is still unverified.
>
> In regards to the benefits this new method provides to online large-scale active learning experiments, paragraph 2 of the discussion uses our theoretical results to explain how larger initial datasets and acquisition sizes benefited our method over original core-sets, as seen in Figure 6 compared to Figure 5.
>
> Other questions you asked:
>
> Does model uncertainty come from the new model or fine-tuned VGG-16?
> We only use one model, which is the fine-tuned VGG-16, for both vector embeddings (for computing distances in core-sets) and confidences.
>
> Clarify why this new method has no additional runtime over the greedy approach.
> We use the output embeddings to compute distances, then use the confidences from the default linear-complexity transformation of VGG-16's last layer on these embeddings, which is not the bottleneck computation.

---

> > ### Comment · Reviewer_FMtn · 2021-11-29
> > **Appreciate the clarity and answers.**
> >
> > Thank you to the authors for providing answers to a number of my questions. My concerns (re experimental results and others) remain given that there have been no further addition to the paper, and retain my score.

---

### Official Review · Reviewer_qzud · 2021-11-03

**Correctness:** 4
**Technical Novelty And Significance:** 3
**Empirical Novelty And Significance:** 3
**Recommendation:** 8
**Confidence:** 4

**Main Review:**

Overall, the paper is well-written, where the authors provide a clear motivation of the proposed scheme, followed by a logical presentation of an enhanced core-set algorithm based on "uncertainty-scaled" distances, supported by experimental results that show the potential advantages of the proposed approach.

Despite the simplicity of the proposed scheme, the authors show that the use of doubt/confidence to scale the distances in evaluating the data points can significantly improve the performance of the original greedy core-set algorithm.
The authors provide interesting insights (e.g., in Fig. 3) on how scaling the distances using confidence/doubt can enhance sampling efficiency, and also discuss when such scaling may (or may not) be beneficial (e.g., in Fig. 7).

One major concern about the current work is that it only considers the original greedy core-set algorithm for comparison, although there has been a large number of algorithms that have been spawned by this original core-set scheme.
It would be interesting and also important to compare the performance of the proposed uncertainty-scaled core-set algorithm with other core-set algorithms, in order to demonstrate the proposed algorithm meaningful advances the current state-of-the-art.

Of special interest is the comparison against recent Bayesian core-set algorithms, which are capable of naturally incorporating uncertainties in the predictions into the selection scheme based on a Bayesian paradigm.

Currently, it appears that the doubt only considers the prediction error although there may be different ways for quantifying this doubt/confidence/uncertainty.
For example, we may consider Bayesian beliefs that may be continuously updated during the selection process, and it may be possible to consider the variance of the prediction error (or its confidence interval).
It would be meaningful to provide some additional justifications of the choice made in the proposed algorithm and discuss limitations (if any) of the current scheme, as well as alternative ways for quantifying uncertainty and their potential pros/cons.





**Summary Of The Paper:**

In this paper, the authors enhance the original core-set algorithm proposed in Sener & Savarese (2018) by incorporating distance measures that are weighted by confidence/uncertainty levels.
Based on CIFAR10/100 and SVHN image classification benchmarks, the authors show that the proposed "doubt-weighted" core-set algorithm can improve the active learning performance compared to the original greedy core-set algorithm.


**Summary Of The Review:**

This paper presents an extension of the original greedy core-set algorithm.
The proposed scheme is relatively simple yet well-motivated, and the experimental results show that uncertainty-based distance scaling can improve the sampling efficiency of the resulting active learning scheme.
However, the authors compare the performance of the proposed algorithm only to the original core-set algorithm, although the original algorithm has spawned a large number of more recent algorithms that have been shown to improve performance.
Comparison with more recent coreset algorithms - especially, Bayesian coreset algorithms that naturally incorporate uncertainties into their predictions - would strengthen the paper by demonstrating its potential advantage against the current state-of-the-art.
Furthermore, it would be beneficial to include further discussions about alternative ways for assessing doubt/confidence/uncertainty and their pros/cons as well as any limitations of the current approach (of estimating doubt).

---

> ### Author Response · Authors · 2021-11-23
> **Thank you for the feedback**
>
> Your comments on Bayesian core-sets and alternatives to doubt agree with the other reviews - indeed these are areas we should investigate to improve the strength of our work.

---

### Official Review · Reviewer_Y5dH · 2021-11-03

**Correctness:** 3
**Technical Novelty And Significance:** 2
**Empirical Novelty And Significance:** 2
**Recommendation:** 3
**Confidence:** 4

**Main Review:**

Strength
-	The authors propose a straight-forward way to incorporate uncertainty into Coreset, which is a reasonable and natural extension to Coreset.
-	The authors provide thorough theoretical analysis of potential rationales behind the improved performance.
-	Empirical results suggest the proposed method outperform the vanilla coreset solution, and the ablation study informs us that both beam search and uncertainty weighting is needed to achieve good performance and low variance.

Weakness
-	The paper seems to be not ready enough with the lack of content. Specifically, there is significant lack of literature reviews of the recent advances in active learning communities, especially in the family of Bayesian learning (e.g.  Bayesian Coreset is directly relevant to this work, which is ignored in the literature review) which usually provide much more reliable uncertainty measurements then ‘doubt’. In addition, there has been multiple works discussing similar direction (combining uncertainty with Coreset) such as [1][2], it might also worth discussing how the current setup is more advanced than related studies.
-	Another problem is the lack of comparison to recent SOTA baselines. Despite that both BADGE and BatchBALD are mentioned in related work section, the authors only compare to vanilla Coreset which is rather old and non-SOTA. The reasoning that “original core-set algorithm improved significantly from those baselines, we expect improvement over the original core-set algorithm to imply similar or greater improvement as well” does not make sense as the paper was published quite early and only compared to less advanced algorithms at the time.
-	The key assumption that “doubt acts as a cheap but noisy estimate of the distance to the nearest point with zero error” is a bit questionable to this reviewer, as it is known that neural networks tend to be over-confident in some regions despite the error is high. Some discussions on when the assumption might break and perhaps the consequence in that scenario would be appreciated.

Reference
[1] Confident Coreset for Active Learning in Medical Image Analysis, Kim et al
[2] Bayesian Active Learning by Disagreements: A Geometric Perspective, Cao et al


**Summary Of The Paper:**

In this paper, the authors propose to improve the vanilla greedy Core-set active learning algorithm by (1) weighting the distance with uncertainty (measured by doubt, $1-\max_yP(y|x)$) and (2) use beam search instead of greedy search where the beams are selected by average uncertainty. They show with several toy examples that this way the samples are concentrated closer to low-confidence region. They further try to find theoretical groundings for the advantage of the proposed algorithm with several assumptions. Finally they show that the proposed algorithm outperform vanilla coreset on CIFAR and  SVHN, and provide some ablation studies to showcase the effect of beam search and uncertainty weighting.

**Summary Of The Review:**

The authors propose an extension to the vanilla Coreset which is novel but relatively straight-forward. They provide some theoretical grounding for the designed algorithm, and show with some preliminary result. However due to the obvious lack of comparison to SOTA and insufficient discussion on related work, this reviewer believe that the work is still premature to be accepted.

---

> ### Author Response · Authors · 2021-11-23
> **We appreciate your thorough review**
>
> Thank you for your constructive feedback. We will aim to improve the coverage of our literature review, breadth of experimentation baselines and provide further discussion on our key assumptions for the theoretical results.
>
> Regarding the lack of comparison to recent SOTA, our original intention was to show a simple improvement to greedy core-sets that incurs no cost to computational complexity.

---

### Decision · Program_Chairs · 2022-01-20

**Decision:**

Reject

**Comment:**

The paper presents an improvement to the core-set active learning algorithm by leveraging distance measures weighted by uncertainty scores and using beam search instead of greedy search.

The reviewers agreed that the paper provides a nice theoretical analysis as well as motivation for the proposal, as well an ablation that shows the proposal indeed empirically outperforms the original core-set algorithm. However, the reviewers also agreed that additional important comparisons would make the paper more convincing, including Bayesian core-set algorithms as well as other recent proposals based on the original core-set algorithm.